# VPP: Efficient Conditional 3D Generation via Voxel-Point Progressive Representation

**Zekun Qi** [†]    **Muzhou Yu** [†]    **Runpei Dong** [‡]

Xi'an Jiaotong University

{qzk139318, muzhou9999, runpei.dong}@stu.xjtu.edu.cn

**Kaisheng Ma** [¶]

Tsinghua University

kaisheng@mail.tsinghua.edu.cn

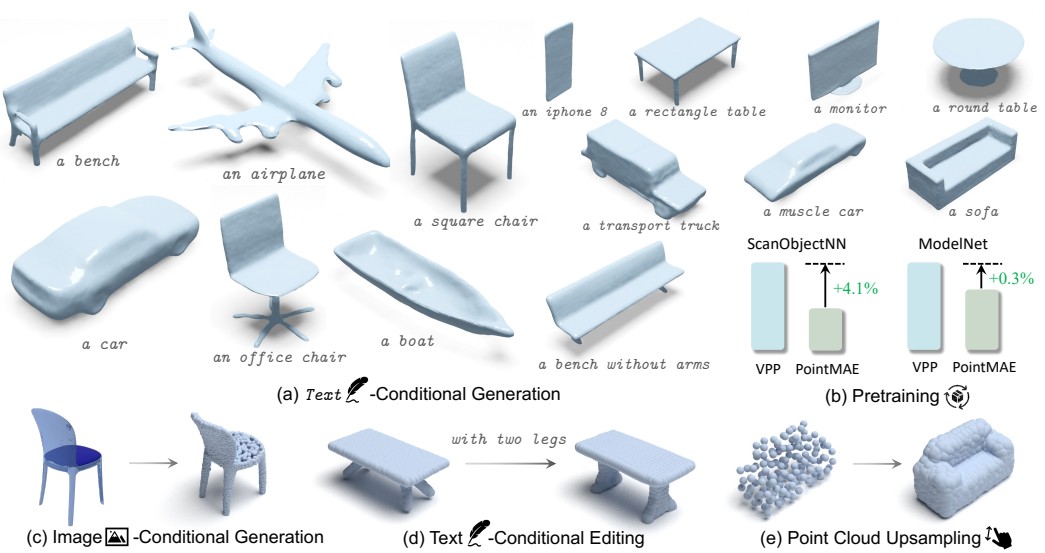

(a) *Text* ✎ -Conditional Generation

(b) Pretraining 🔄

(c) Image 🖼 -Conditional Generation    (d) Text ✎ -Conditional Editing    (e) Point Cloud Upsampling 🔼

Figure 1: Overview of VPP's applications through generative modeling.

## Abstract

Conditional 3D generation is undergoing a significant advancement, enabling the free creation of 3D content from inputs such as text or 2D images. However, previous approaches have suffered from low inference efficiency, limited generation categories, and restricted downstream applications. In this work, we revisit the impact of different 3D representations on generation quality and efficiency. We propose a progressive generation method through Voxel-Point Progressive Representation (VPP). VPP leverages structured voxel representation in the proposed Voxel Semantic Generator and the sparsity of unstructured point representation in the Point Upsampler, enabling efficient generation of multi-category objects. VPP can generate high-quality 8K point clouds *within 0.2 seconds*. Additionally, the masked generation Transformer allows for various 3D downstream tasks, such as generation, editing, completion, and pre-training. Extensive experiments demonstrate that VPP efficiently generates high-fidelity and diverse 3D shapes across different categories, while also exhibiting excellent representation transfer performance. Codes will be released at https://github.com/qizekun/VPP.

---

[†]Equal contribution. [‡]Project lead. [¶]Corresponding author.

37th Conference on Neural Information Processing Systems (NeurIPS 2023).

Table 1: Comparison of text-conditioned 3D generation methods on efficiency and applications.

| Method | Latency | Device | Generation Category | Downstream Task |
|--------|---------|--------|---------------------|-----------------|
| DreamFields [37] | 1.2h | 8×TPUv4 | Multi-Category | Generation |
| DreamFusion [65] | 1.5h | 4×TPUv4 | Multi-Category | Generation |
| CLIP-Mesh [58] | 30min | V100 | Multi-Category | Generation |
| CLIP-Sculptor [77] | 0.9s | V100 | Multi-Category | Generation |
| Point·E (40M) [59] | 25s | V100 | Multi-Category | Generation |
| ShapeCrafter [24] | - | - | Single-Category | Generation, Editing |
| LION [99] | 27s | V100 | Single-Category | Generation, Completion |
| SDFusion [9] | - | - | Single-Category | Generation, Completion |
| **VPP (Ours)** | **0.2s** | **RTX 2080Ti** | **Multi-Category** | **Generation, Editing, Completion, Pretraining** |

# 1   Introduction

In the last few years, text-conditional image generation [62, 71, 72, 74] has made significant progress in terms of generation quality and diversity and has been successfully applied in games [7], content creation [47, 48] and human-AI collaborative design [36]. The success of text-to-image generation is largely owed to the large-scale image-text paired datasets [78]. Recently, there has been an increasing trend to boost conditional generation from 2D images to 3D shapes. Many works attempt to use existing text-3D shape paired datasets [7, 9, 24, 50] to train the models, but the insufficient pairs limit these methods to generate multi-category, diverse, and high-res 3D shapes.

To tackle this challenge, some works [37, 65, 82, 94] use NeRF [56] and weak supervision of large-scale vision-language models [70] to generate 3D shapes. Although creative quality is achieved, the optimization costs of differentiable rendering are quite expensive and impractical. Some methods use images as bridges [38, 59] for the text-conditional 3D generation, which improves the quality-time trade-off. However, the complex multi-stage model leads to long inference latency and cumulative bias. Moreover, most methods [58, 76] are task-limited due to specific model design that can not be implemented to more downstream tasks, *e.g.*, editing, leading to narrow application scope. To sum up, current methods are faced with three challenges: *low inference efficiency*, *limited generation category*, and *restricted downstream tasks*, which have not been solved by a unified method.

One of the most crucial components in 3D generation is the *geometry representation*, and mainstream methods are typically built with voxel, 3D point cloud, or Signed Distance Function (SDF) [11, 61]. However, the aforementioned issues in 3D generation may come from the different properties of these representations, which are summarized as follows:

- **Voxel** is a structured and explicit representation. It shares a similar form with 2D pixels, facilitating the adoption of various image generation methods. However, the dense nature of voxels results in increased computational resources and time requirements when generating high-resolution shapes.

- **3D Point Cloud** is the unordered collection of points that explicitly represents positional information with sparse semantics, enabling a flexible and efficient shape representation. However, 3D point clouds typically dedicatedly designed architectures, *e.g.*, PointNet [66, 67].

- **SDF** describes shapes through an implicit distance function. It is a continuous representation method and is capable of generating shapes with more intricate details than points or voxels. But it requires substantial computation for high-resolution sampling and is sensitive to initialization.

Due to significant shape differences across various categories, the structured and explicit positional information provides spatial cues, thereby aiding the generalization of multiple categories. It coincides with previous multi-category generation methods [76, 77]. In contrast, single-category generation may require less variation. Hence an implicit and continuous representation would contribute to the precise expression of local details [9, 25].

Motivated by the analysis above, we aim to leverage the advantage of different forms of representation. Therefore, a novel **V**oxel-**P**oint **P**rogressive (**VPP**⚡) Representation method is proposed to achieve efficient and universal 3D generation. We use voxel and point as representations for the coarse to fine stages to adapt to the characteristics of 3D. To further enhance the inference speed, we employ mask generative Transformers for parallel inference [5] in both stages, which can concurrently obtain broad applicability scope [42]. Based on voxel VQGAN [21], we utilize the discrete semantic codebooks as the reconstruction targets to obtain authentic and rich semantics. Given the substantial computational resources required by high-resolution voxels, we employ sparse points as representations in the

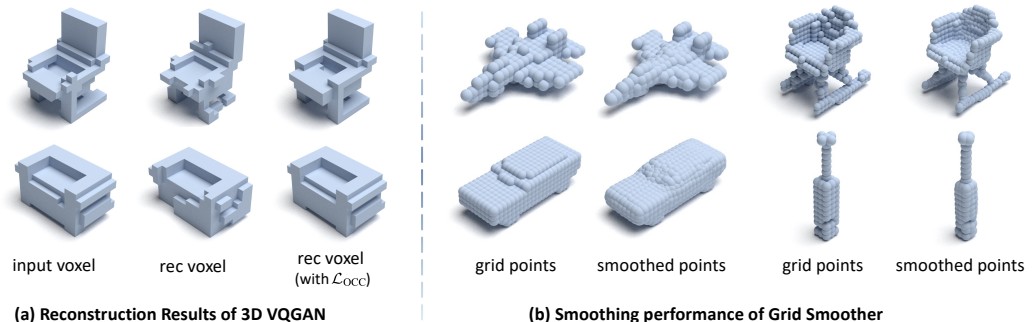

| input voxel | rec voxel | rec voxel (with $\mathcal{L}_{OCC}$) | | grid points | smoothed points | grid points | smoothed points |

**(a) Reconstruction Results of 3D VQGAN** | **(b) Smoothing performance of Grid Smoother**

Figure 2: Qualitative examples. (a) 3D VQGAN reconstruction results with or without $\mathcal{L}_{OCC}$ (Section 2.1). (b) 3D point clouds result with or without proposed Grid Smoother (Section 2.2).

second stage. Inspired by the masked point modeling approaches based on position cues [60, 98], we utilize the voxels generated in the first stage as the positional encoding for generative modeling. We reconstruct the points and predict semantic tokens to obtain sparse representations. Table 1 compares the efficiency and universality of text-conditional 3D generation methods.

In summary, our contributions are: (1) We propose a novel voxel-point progressive generation method that shares the merits of different geometric representations, enabling efficient and multi-category 3D generation. Notably, VPP is capable of generating high-quality 8K point clouds **within 0.2 seconds** on a single RTX 2080Ti. (2) To accommodate 3D generation, we implement unique module designs, *e.g.* 3D VQGAN, and propose a Central Radiative Temperature Schedule strategy in inference. (3) As a universal method, VPP can complete multiple tasks such as editing, completion, and even pre-training. To the best of our knowledge, VPP is the first work that unifies various 3D tasks.

## 2 VPP

Our goal is to design a 3D generation method by sharing the merits of both geometric representations, which is capable of addressing the three major issues encountered in previous generation methods. In this work, we propose VPP, a Voxel-Point Progressive Representation method for two-stage 3D generation. To tackle the problem of *limited generation categories*, VPP employs explicit voxels and discrete VQGAN for semantics representation in the first stage. To address the issue of *low inference efficiency*, VPP employs efficient point clouds as representations in the second stage and leverages the parallel generation capability of the mask Transformer. Given the wide range of applications of generative modeling, VPP resolves the problem of *restricted downstream tasks*. The training overview of VPP is shown in Fig. 3.

### 2.1 How Does 2D VQGAN Adapt to 3D Voxels?

Due to the formal similarity between voxels and pixels, we can readily draw upon various image generation methods. VQGAN [21] is capable of generating realistic images through discrete semantic codebooks and seamlessly integrating with mask generative Transformers [5, 6]. We utilize 3D volumetric SE-ResNet [35] as the backbone of VQGAN encoder $E$, decoder $G$ and discriminator $D$.

Given one voxel observation $\mathbf{v} \in \mathbb{R}^3$ whose occupancy is defined as function $o : \mathbb{R}^3 \to [0, 1]$. The original VQGAN is trained by optimizing a vectorized quantized codebook with the loss $\mathcal{L}_{VQ}$ during autoencoding while training a discriminator between real and reconstructed voxels with loss $\mathcal{L}_{GAN}$. Denoting the reconstructed voxel result as $\hat{\mathbf{v}}$, the overall loss of vanilla VQGAN can be written as:

$$\underbrace{\|\mathbf{v} - \hat{\mathbf{v}}\|^2 + \|sg[E(\mathbf{v})] - z_{\mathbf{q}}\|_2^2 + \|sg[z_{\mathbf{q}}] - E(\mathbf{v})\|_2^2}_{\mathcal{L}_{VQ}(E, G, \mathcal{Z})} + \underbrace{\left[\log D(\mathbf{v}) + \log(1 - D(\hat{\mathbf{v}}))\right]}_{\mathcal{L}_{GAN}(\{E, G, \mathcal{Z}\}, D)}.$$

However, unlike 2D images, 3D voxels only have non-zero values in the region that encompasses the object in grid centroids. This may lead to a short-cut solution that degrades by outputting voxels with values as close to zero as possible, *i.e.*, all empty occupancy. To address this issue, we propose to

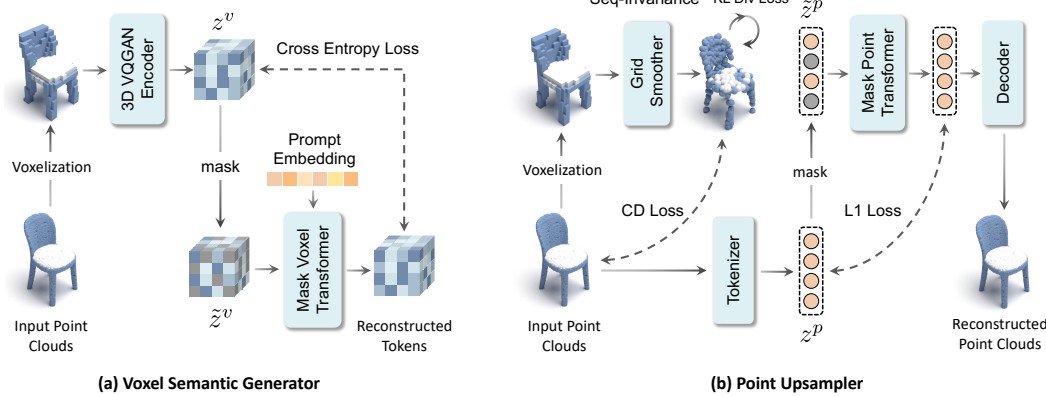



**(a) Voxel Semantic Generator**        **(b) Point Upsampler**



Figure 3: Training process overview of VPP. (a) In the coarse stage, the VQ codebook is reconstructed via Mask Voxel Transformer conditioned on prompt embeddings. (b) In the fine stage, sparse semantic tokens are reconstructed via the Mask Point Transformer conditioned on positional embeddings.

minimize the occupancy disparity between the reconstructed and ground truth voxels. Formally, let $\zeta \circ o(\mathbf{v})$ be the averaged occupancy in SE(3) space, the overall loss $\mathcal{L}_{\text{3D-VQGAN}}$ is:

$$\mathcal{L}_{\text{3D-VQGAN}} = \mathcal{L}_{\text{VQ}} + \mathcal{L}_{\text{GAN}} + \underbrace{\|\zeta \circ o(\mathbf{v}) - \zeta \circ o(\hat{\mathbf{v}})\|_2^2}_{\mathcal{L}_{\text{OCC}}}, \tag{1}$$

where $\mathcal{L}_{\text{OCC}}$ is proposed to minimize the occupancy rate for better geometry preservation.

## 2.2  How to Bridge the Representation Gap Between Voxels and Points?

We aim to utilize the voxel representation as the intermediate coarse generation before fine-grained point cloud results. However, voxels are uniform and discrete grid data, while point clouds are unstructured with unlimited resolutions, which allows for higher density in complex structures. Besides, point clouds typically model the surfaces, while voxel representation is solid in nature. This inevitably leads to a *representation gap* issue, and a smooth transformation that bridges the representation gap is needed. We employ the Lewiner algorithm [40] to extract the surface of the object based on voxel boundary values and utilize a lightweight sequence-invariant Transformer [84] network $\mathcal{T}_\eta$ parametrized with $\eta$ as *Grid Smoother* to address the issue of grid discontinuity. Specifically, given generated point clouds $\mathcal{P} = \{\mathbf{p}_i \in \mathbb{R}^3\}_{i=1}^N$ with $N$ points. The model is trained by minimizing the geometry distance between generated and ground truth point clouds $\mathcal{G} = \{\mathbf{g}_i \in \mathbb{R}^3\}_{i=1}^N$ that is downsampled to $N$ with farthest point sampling (FPS). To make the generated surface points more uniformly distributed, we propose $\mathcal{L}_{\text{uniform}}$ that utilizes the Kullback-Leibler (KL) divergence $D_{\text{KL}}$ between the averaged distance of every centroid $\mathbf{p}_i$ and its K-Nearest Neighborhood (KNN) $\mathcal{N}_i$. Then we optimize the parameter $\eta$ of Grid Smoother by minimizing loss $\mathcal{L}_{\text{SMO}}$:

$$\mathcal{L}_{\text{SMO}} = \underbrace{\frac{1}{|\mathcal{T}_\eta(\mathcal{P})|} \sum_{\mathbf{p} \in \mathcal{T}_\eta(\mathcal{P})} \min_{\mathbf{g} \in \mathcal{G}} \|\mathbf{p} - \mathbf{g}\| + \frac{1}{|\mathcal{G}|} \sum_{\mathbf{g} \in \mathcal{G}} \min_{\mathbf{p} \in \mathcal{T}_\eta(\mathcal{P})} \|\mathbf{g} - \mathbf{p}\|}_{\mathcal{L}_{\text{CD}}} + D_{\text{KL}} \underbrace{\left[ \sum_{j \in \mathcal{N}_i} \|\mathbf{p}_i - \mathbf{p}_j\|_2, \mathcal{U} \right]}_{\mathcal{L}_{\text{uniform}}}, \tag{2}$$

where $\mathcal{L}_{CD}$ is $\ell_1$ Chamfer Distance [23] geometry disparity loss, $\mathcal{U}$ is the uniform distribution.

## 2.3  Voxel-Point Progressive Generation Trough Mask Generative Transformer

We employ mask generative Transformers [5] for rich semantic learning in both the voxel and point stages. Mask generative Transformer can be seen as a special form of denoising autoencoder [34, 85]. It forces the model to reconstruct masked input to learn the statistical correlations between local patches. Furthermore, generating reconstruction targets through a powerful tokenizer can further enhance performance [2, 19, 91, 104]. From the perspective of discrete variational autoencoder

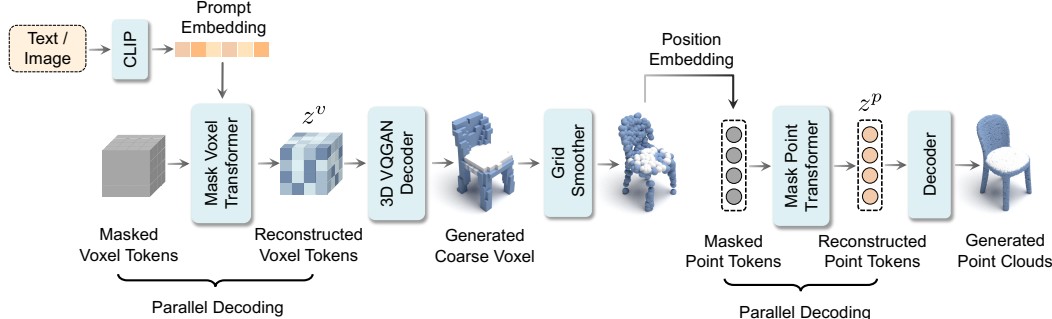

Figure 4: Inference process overview of VPP. Point clouds are generated conditioned on the input text or images. Parallel decoding is adopted for efficient Mask Transformer token prediction.

(dVAE) [2, 26, 39, 71], the overall optimization is to maximize the *evidence lower bound* (ELBO) [17, 32, 73] of the log-likelihood $\mathrm{P}(x_i|\tilde{x}_i)$. Let $x$ denote the original data, $\tilde{x}$ the masked data, and $z$ the semantic tokens, the generative modeling can be described as:

$$\sum_{(x_i,\tilde{x}_i)\in\mathcal{D}} \log \mathrm{P}(x_i|\tilde{x}_i) \geq \sum_{(x_i,\tilde{x}_i)\in\mathcal{D}} \Big( \mathbb{E}_{z_i\sim\mathrm{Q}_\phi(\mathbf{z}|x_i)}\big[\log \mathrm{P}_\psi(x_i|z_i)\big] - D_{\mathrm{KL}}\big[\mathrm{Q}_\phi(\mathbf{z}|x_i), \mathrm{P}_\theta(\mathbf{z}|\tilde{z}_i)\big]\Big), \quad (3)$$

where (1) $\mathrm{Q}_\phi(z|x)$ denotes the discrete semantic tokenizer; (2) $\mathrm{P}_\psi(x|z)$ is the tokenizer decoder to recover origin data; (3) $\tilde{z} = \mathrm{Q}_\phi(z|\tilde{x})$ denotes the masked semantic tokens from masked data; (4) $\mathrm{P}_\theta(z|\tilde{z})$ reconstructs masked semantic tokens in an autoencoding way. In the following sections, we extend the mask generative Transformer to voxel and point representations, respectively. The semantic tokens of voxel and point are described as $z^v$ and $z^p$ in Figs. 3 and 4.

**Voxel Semantic Generator**    We generate semantic voxels with low resolution in the first stage to balance efficiency and fidelity. We utilize *Mask Voxel Transformer* for masked voxel inputs, where the 3D VQGAN proposed in Section 2.1 is used as the tokenizer to generate discrete semantic tokens. Following prior arts [5, 6], we adopt Cosine Scheduling to simulate the different stages of generation. The masking ratio is sampled by the truncated arccos distribution with density function $p(r) = \frac{2}{\pi}(1-r^2)^{-\frac{1}{2}}$. The prompt embedding comes from the CLIP [70] features of cross-modal information. We render multi-view images from the 3D object [68] and utilize BLIP [41] to get the text descriptions of the rendered images. Furthermore, in order to mitigate the issue of excessive dependence on prompts caused by the high masking ratio (*e.g.*, the same results will be generated by one prompt), we introduce *Classifier Free Guidance* [33] (CFG) to strike a balance between diversity and quality. Following Sanghi *et al.* [77], the prompt embeddings are added with Gaussian noise to reduce the degree of confidence. We use $\epsilon \sim \mathcal{N}(0,1)$ and $\gamma \in (0,1)$ as the level of noise perturbation for the trade-off control, which will later be studied.

**Point Upsampler**    Point clouds are unordered and cannot be divided into regular grids like pixels or voxels. Typically, previous masked generative methods [19, 60] utilize FPS to determine the center of local patches, followed by KNN to obtain neighboring points as the geometric structure. Finally, a lightweight PointNet [66, 67] or DGCNN [89] is utilized for patch embedding. However, these methods rely on pre-generated Positional Encodings (PE), rendering the use of fixed positional cues *infeasible* in conditional 3D generation since only conditions like texts are given. To tackle this issue, we adopt the first-stage generated voxels-smoothed point clouds as the PEs. A pretrained Point-MAE [60] encoder is used as the tokenizer (*i.e.*, teacher [19]). The Mask Point Transformer learns to reconstruct tokenizer features, which will be used for the pretrained decoder reconstruction.

## 2.4   Inference Stage

**Parallel Decoding**    An essential factor contributing to the efficient generation of VPP is the parallel generation during the inference stage. The model takes fully masked voxel tokens as input and generates semantic voxels conditioned on the CLIP [70] features of text or image prompts. Following [5, 6], a cosine schedule is employed to select the fixed fraction of the highest confidence masked tokens for

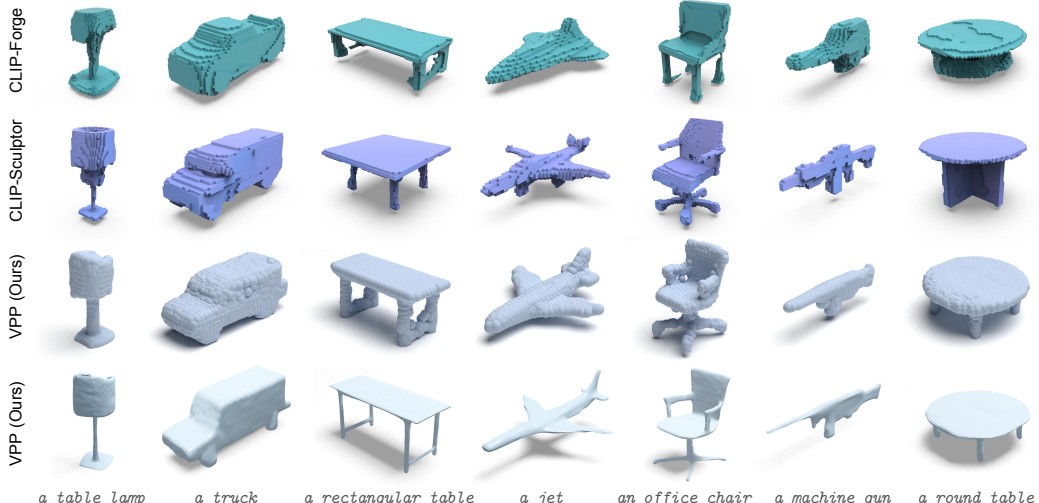

Figure 5: Qualitative comparison of VPP, CLIP-Sculptor [77] and CLIP-Forge [76] on text conditioned 3D generation. VPP generates shapes of higher fidelity with smoother surfaces while being consistent with the input text. More diverse results are shown in Appendix C.

prediction at each step. Furthermore, we pass the generated voxels through Grid Smoother to obtain smoothed point clouds, which will serve as the positional encoding for the Point Upsampler in the upsampling process. Note that all the stages are performed in parallel. The inference overview of VPP is illustrated in Fig. 4.

**Central Radiative Temperature Schedule**  Due to the fact that 3D shape voxels only have non-zero values in the region that encompasses the object in the center, we hope to encourage diversity in central voxels while suppressing diversity in edge vacant voxels. Consequently, we propose the *Central Radiative Temperature Schedule* to accommodate the characteristics of 3D generation. As for the voxel with a size of $R \times R \times R$, we calculate the distance $r$ to the voxel center for each predicted grid. Then, we set the temperature for each grid as $T = 1 - (r/R)^2$, where $T$ represents the temperature. Thus, we can achieve the grid that is closer to the center will have a higher temperature to encourage more diversity and vice versa.

## 3   Experiments

### 3.1   Conditional 3D Generation

**Text-conditioned Generation**  Following [59, 76, 77], we use Accuracy (ACC), Fréchet Inception Distance (FID) [31], and Inception Score (IS) [75] as metrics to assess the generation quality and diversity. We follow [76] to define 234 predetermined text queries and apply a classifier for evaluation. For a fair comparison, all methods use "a/an" as the prefix for text prompts. Similar to [59], we employ a double-width PointNet++ model [67] to extract features and predict class probabilities for point clouds in ShapetNet [4]. Table 2 shows the comparison between VPP and other methods. It can be observed that our VPP outperforms all other methods substantially in all metrics. Notably, the FID metric of ours is much lower than other methods, demonstrating that our method generates higher quality and richer diversity of 3D shapes.

By employing a *Shape As Points* [52, 63] model pre-trained on multi-category ShapeNet data, VPP is capable of performing smooth surface reconstruction on generated point clouds. In Blender rendering, we employ varnish and automatic smoothing techniques to improve the display quality of the generated mesh. We qualitatively show in Fig. 5 that our method can generate higher-quality shapes and remain faithful to text-shape correspondence. It is noteworthy that VPP can generate smoother 3D shapes, *e.g.*, the truck. Besides, our generation results are visually pleasing across different object categories with complex text descriptions. Therefore, the above advantages enable our method to be much more useful in practical applications.

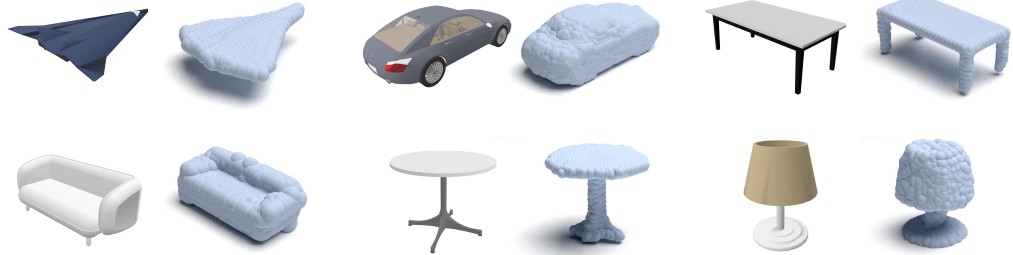

Figure 6: Qualitative examples of single image conditioned 3D generation with VPP.

Table 2: Text conditioned 3D generation results on ShapeNetCore13 [4] dataset. Point-E [59] is pre-trained on a large-scale private point-text paired dataset containing millions of samples, and we report its zero-shot transfer results on ShapeNet. Accuracy (ACC), Fréchet Inception Distance (FID), and Inception Score (IS) are reported.

| Method | ACC↑ | FID↓ | IS↑ |
|---|---|---|---|
| CLIP-Forge [76] | 83.33% | 2425.25 | - |
| CLIP-Sculptor [77] | 87.08% | 1480.11 | - |
| Point-E (40M text-only) [59] | 62.56% | 85.37 | 7.93 |
| VPP (ours) | **88.04%** | **29.82** | **10.64** |

**Image-Conditioned Generation** By leveraging the CLIP features of 2D images as conditions in the Voxel Semantic Generator, VPP also possesses the capability of single-view reconstruction. Fig. 6 illustrates image-conditional generated samples of VPP. It can be observed that the generated point clouds are of detailed quality and visually consistent with the input image. For example, the flat wings of an airplane and the curved backrest of a chair.

## 3.2 Text-Conditioned 3D Editing and Upsampling

Text-conditioned 3D editing and upsample completion tasks are more challenging than most conditional 3D generation tasks, which most of previous methods can not realize due to the inflexible model nature. In contrast, our VPP can be used for a variety of shape editing and upsample completion without extra training or model fine-tuning.

**Text-Conditioned 3D Editing** Since we use the masked generative Transformer in VPP, our model can be conditioned on any set of point tokens: we first obtain a set of input point tokens, then mask the tokens corresponding to a local area, and decode the masked tokens with unmasked tokens and text prompts as conditions. We show the examples in Fig. 7 (a). The figure provides examples of VPP being capable of changing part attributes (view) and modifying local details (view) of the shape, which correctly corresponds to the text prompts.

**Point Cloud Upsampling** Except for editing, we also present our conditional generative model for the point cloud upsample completion, where the sparse point cloud inputs are completed as a dense output. We use the second stage Point Upsampler to achieve this, and the results are illustrated in Fig. 7 (b). It is observed that VPP generates completed and rich shapes of high fidelity while being consistent with the input sparse geometry.

## 3.3 Transfer Learning on Downstream Tasks

After training, the Mask Transformer has learned powerful geometric knowledge through generative reconstruction [30], which enables VPP to serve as a self-supervised learning method for downstream representation transferring. Following previous works [19, 60, 68], we fine-tune the Mask Point Transformer encoder, *i.e.*, the second-stage *Point Upsampler* for 3D shape recognition.

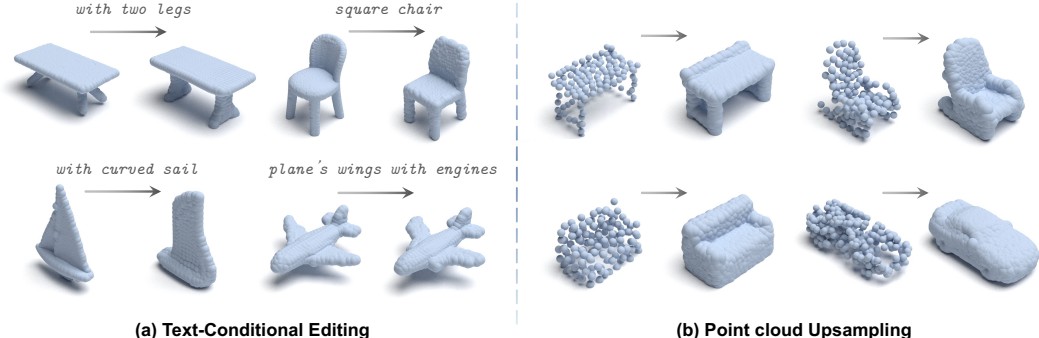

| (a) Text-Conditional Editing | (b) Point cloud Upsampling |

Figure 7: Qualitative examples of text conditioned 3D editing and upsample completion with VPP.

Table 3: Downstream 3D object classification results on the ScanObjectNN and ModelNet40 datasets. The inference model parameters #P (M), FLOPS #F (G), and overall accuracy (%) are reported.

| Method | #P | #F | ScanObjectNN | | | ModelNet40 | |
| --- | --- | --- | --- | --- | --- | --- | --- |
| | | | OBJ_BG | OBJ_ONLY | PB_T50_RS | 1k P | 8k P |
| *Supervised Learning Only* | | | | | | | |
| PointNet [66] | 3.5 | 0.5 | 73.3 | 79.2 | 68.0 | 89.2 | 90.8 |
| PointNet++ [67] | 1.5 | 1.7 | 82.3 | 84.3 | 77.9 | 90.7 | 91.9 |
| DGCNN [89] | 1.8 | 2.4 | 82.8 | 86.2 | 78.1 | 92.9 | - |
| PointCNN [43] | 0.6 | - | 86.1 | 85.5 | 78.5 | 92.2 | - |
| PCT [27] | 2.88 | 2.3 | - | - | - | 93.2 | - |
| PointMLP [53] | 12.6 | 31.4 | - | - | 85.4±0.3 | 94.5 | - |
| PointNeXt [69] | 1.4 | 3.6 | - | - | 87.7±0.4 | 94.0 | - |
| *with Self-Supervised Representation Learning* | | | | | | | |
| Transformer [84] | 22.1 | 4.8 | 83.04 | 84.06 | 79.11 | 91.4 | 91.8 |
| Point-BERT [98] | 22.1 | 4.8 | 87.43 | 88.12 | 83.07 | 93.2 | 93.8 |
| Point-MAE [60] | 22.1 | 4.8 | 90.02 | 88.29 | 85.18 | 93.8 | 94.0 |
| Point-M2AE [102] | 14.8 | 3.6 | 91.22 | 88.81 | 86.43 | 94.0 | - |
| **VPP w/o vot.** | 22.1 | 4.8 | 92.77 | 91.56 | 88.65 | 93.8 | 94.0 |
| **VPP w/ vot.** | 22.1 | 4.8 | **93.11** | **91.91** | **89.28** | **94.1** | **94.3** |
| *with Pretrained Cross-Modal Teacher Representation Learning* | | | | | | | |
| ACT [19] | 22.1 | 4.8 | 93.29 | 91.91 | 88.21 | 93.7 | 94.0 |
| I2P-MAE [103] | 12.9 | 3.6 | 94.15 | 91.57 | 90.11 | 94.1 | - |
| ReCon [68] | 43.6 | 5.3 | 95.18 | 93.29 | 90.63 | 94.5 | 94.7 |

ScanObjectNN [83] and ModelNet [92] are currently the two most challenging 3D object datasets, which are obtained through real-world sampling and synthesis, respectively. We show the evaluation of 3D shape classification in Table 3. It can be seen that: (i) Compared to any supervised or self-supervised method that only uses point clouds for training, VPP achieves the best generalization performance. (ii) Notably, VPP outperforms Point-MAE by +4.0% and +0.3% accuracy on the most challenging PB_T50_RS and ModelNet40 benchmark.

## 3.4 Diversity & Specific Text-Conditional Generation

We show the diverse qualitative results of VPP on text-conditional 3D generation in Fig. 8 (a). It can be observed that VPP can generate a broad category of shapes with rich diversity while remaining faithful to the provided text descriptions. Meanwhile, we present the qualitative results of VPP on more specific text-conditional 3D generation in Fig. 8 (b). Notably, VPP can generate high-fidelity shapes that react well to very specific text descriptions, like `a cup chair`, `a round table with four legs` and `an aircraft during World War II`, *etc*.

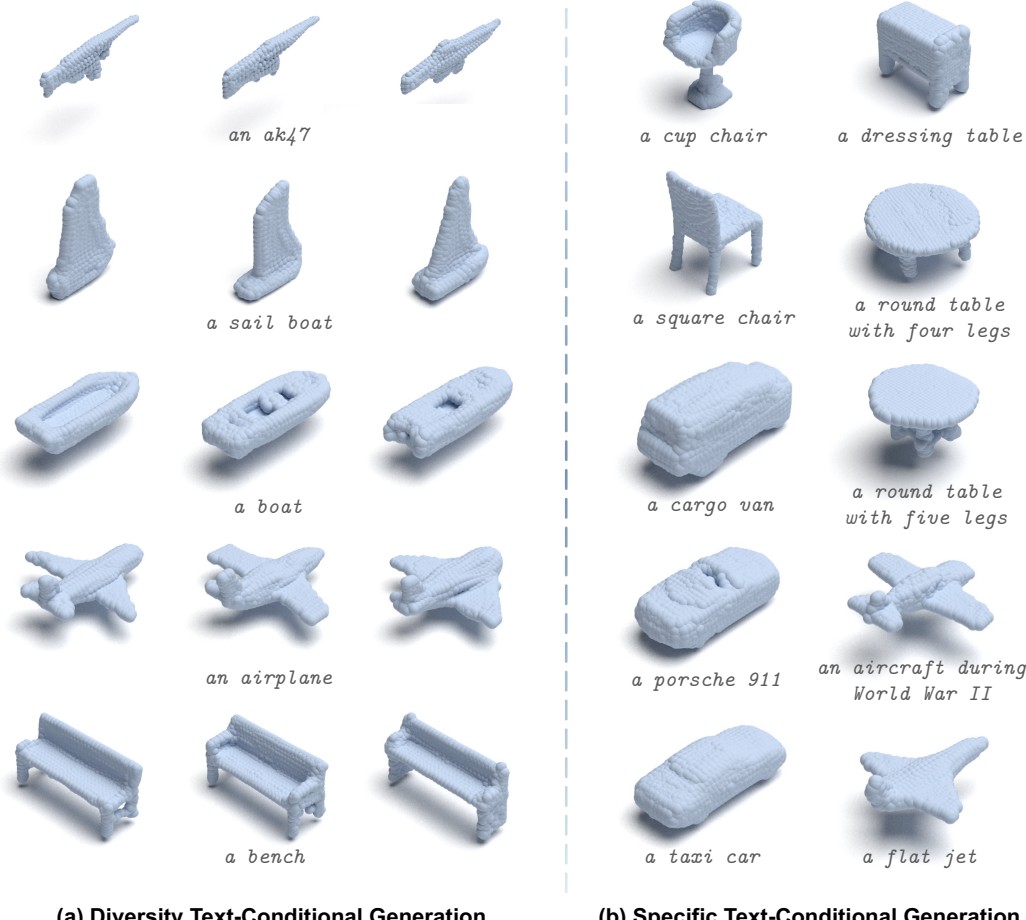

| (a) Diversity Text-Conditional Generation | (b) Specific Text-Conditional Generation |

Figure 8: (a) **Diversity** qualitative results of VPP on text conditioned 3D generation. (b) Qualitative results of VPP on more **specific** text-conditioned 3D generation. VPP can generate a broad category of shapes with rich diversity and high fidelity while remaining faithful to the provided text descriptions. Besides, VPP can react to very specific text descriptions, like `a cup chair`.

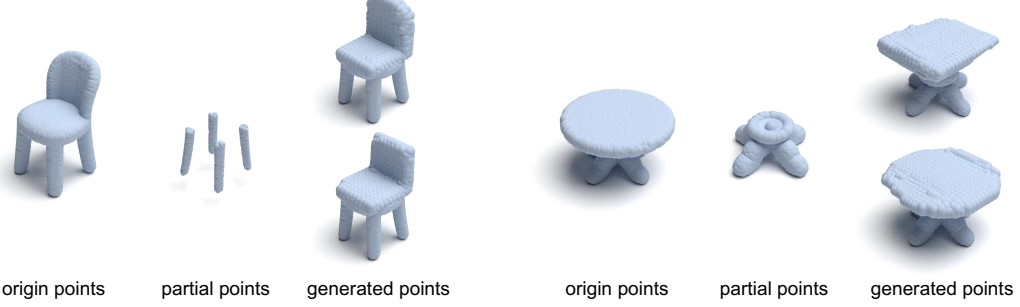

origin points    partial points    generated points      origin points    partial points    generated points

Figure 9: Partial inputs point clouds completion results of VPP. Our model is capable of generating diverse completion samples.

## 3.5 Partial Completion

We present the point cloud partial completion experiments in Fig. 9. By employing a block mask on the original point clouds, VPP can generate partial samples, which illustrates the partial completion capability of VPP. Besides, the generated samples exhibit diversity, further demonstrating the diverse performance of VPP.

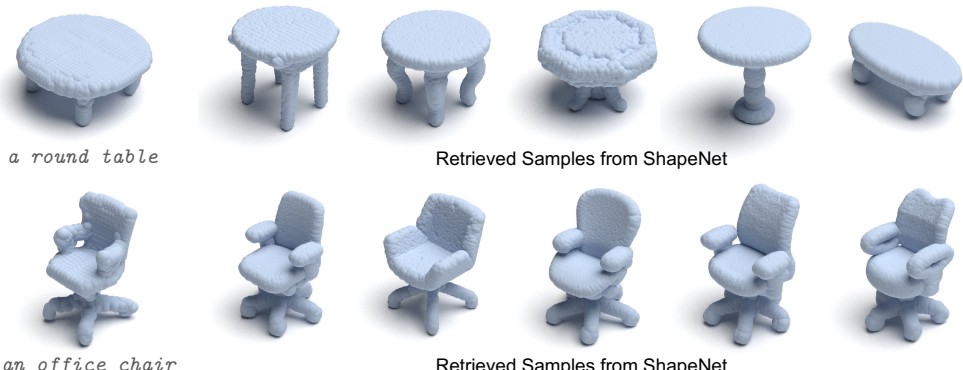

a round table          Retrieved Samples from ShapeNet

an office chair          Retrieved Samples from ShapeNet

Figure 10: ShapeNet training data retrieval ablation experiment.

### 3.6 ShapeNet Retrieval Experiment

We conduct the retrieval evaluation on samples generated by VPP from the ShapeNet dataset. The results are shown in Fig. 10. It can be observed that there are no completely identical samples, proving the great generation ability of VPP is not caused by overfitting the ShapeNet dataset. The VPP results more likely come from the understanding and integration of the learned shape knowledge.

## 4 Related Works

Conditional 3D generation with 2D images or text has witnessed rapid development in recent years. One line of works focuses on solving the ill-posed image-conditioned 3D generation (*i.e.*, single-view reconstruction), where impressive results have been achieved [10, 23, 57, 88]. By introducing a text-shape paired dataset based on ShapeNet [4] that includes chair and table categories, [7] pioneered another line of single-category text-conditioned shape generation. Facilitated with this human-crawled 3D data, ShapeCrafter [24] further enables text-conditioned shape edition in a recursive fashion. To leverage easily rendered 2D images rather than human-crafted languages, several works propose to train rendered image-to-shape generation [9, 51, 76, 77] which enables text-conditioned generation through pretrained large vision-language (VL) models, *i.e.*, CLIP [70]. Besides rendered images, Point-E [59] and Shape-E [38] propose to use real-world text-image-shape triplets from a large-scale in-house dataset, where images are used as the representation bridge.

However, 3D data is significantly lacking and expensive to collect [19]. DreamFields [37] first achieves text-only training by using NeRF [56], where rendered 2D images are weakly supervised to align with text inputs using CLIP. Following this direction, DreamFusion [65] incorporates the distillation loss of a diffusion model [74]. Dream3D [94] improves the knowledge prior by providing the prompt shapes initialization. Though impressive results are obtained, these approaches also introduce significant time costs because of the case-by-case NeRF training during inference. More recently, TAPS3D aligns rendered images and text prompts based on DMTet [79] to enable text-conditional generation, but its generalizability is limited to only four categories. 3DGen [28] achieves high-quality generation through the utilization of Triplane-based diffusion models [3, 80] and the Objaverse dataset [14]. Zero-1-to-3 [46] employs relative camera viewpoints as conditions for diffusion modeling. Based on this, One-2-3-45 [45] achieves rapid open-vocabulary mesh generation through multi-view synthesis. Fantasia3D [8] leverages DMTet to employ differentiable rendering of SDF as a substitute for NeRF, enabling a more direct mesh generation. SJC [87] and ProlificDreamer [90] achieve astonishing results by proposing the variational diffusion distillation.

## 5 Conclusions

In this paper, we present VPP, a novel Voxel-Point Progressive Representation method that achieves efficient and universal conditional 3D generation. We explore the suitability of different 3D geometric representations and design a VQGAN module tailored for 3D generation. Notably, VPP is capable of generating high-quality 8K point clouds in less than 0.2 seconds using a single RTX 2080Ti GPU. Extensive experiments demonstrate that VPP exhibits excellent performance in conditional generation, editing, completion, and pretraining.

## Acknowledgments

This research was supported by the National Key R&D Program of China (2022YFB2804103), the Key Research and Development Program of Shaanxi (2021ZDLGY01-05), the National Natural Science Foundation of China (31970972), and the Institute for Interdisciplinary Information Core Technology (IIISCT).

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

# A  Additional Related Work

3D Representation Learning includes point-based [66, 67], voxel-based [55], and multiview-based methods [29, 81], *etc*. Due to the sparse but geometry-informative representation, point-based methods [20, 69] have become mainstream approaches in object classification [83, 92]. Voxel-based CNN methods [15, 97] provide dense representation and translation invariance, achieving outstanding performance in object detection [12] and segmentation [1, 96]. Furthermore, due to the vigorous development of attention mechanisms [84], 3D Transformers [20, 49, 54] have also brought about effective representations for downstream tasks. Recently, 3D self-supervised representation learning has been widely studied. PointContrast [93] leverages contrastive learning across different views to acquire discriminative 3D scene representations. Point-BERT [98] and Point-MAE [60] first introduce masked modeling pretraining into 3D. ACT [19] pioneers cross-modal geometry understanding via 2D/language foundation models. RECON [68] further proposes to unify generative and contrastive learning. Facilitated by foundation vision-language models like CLIP [70], another line of works are proposed towards open-world 3D representation learning [16, 22, 64, 95, 100, 101].

# B  Implementation Details

## B.1  Experimental Details

**Training Details**  We use ShapeNetCore from ShapeNet [4] as the training dataset. ShapeNet is a clean set of 3D CAD models with rich annotations, including 51K unique 3D models from 55 common object categories. For the acquisition of multi-modal data, we follow ReCon [68] for multi-view rendering and utilize BLIP [41] based on rendered images to obtain textual data. Table 4 shows the training hyperparameters and model architecture information of each part of VPP.

Table 4: Training recipes for 3D VQGAN, Voxel Generator, Grid Smoother and Point Upsampler.

| Config | 3D VQGAN | Voxel Generator | Grid Smoother | Point Upsampler |
|---|---|---|---|---|
| **Training Parameters** | | | | |
| Optimizer | Adam | AdamW | AdamW | AdamW |
| Learning rate | 1e-4 | 1e-3 | 1e-3 | 1e-3 |
| Weight decay | 1e-4 | 5e-2 | 5e-2 | 5e-2 |
| Training epochs | 100 | 100 | 100 | 300 |
| Warmup epochs | - | 5 | 5 | 10 |
| Learning rate scheduler | cosine | cosine | cosine | cosine |
| Batch size | 32 | 128 | 128 | 128 |
| Drop path rate | - | 0.1 | 0.1 | 0.1 |
| Input point size | 8192 | 8192 | 8192 | 1024 |
| **Model Architecture** | | | | |
| Backbone | CNN | Transformer | Transformer | Transformer |
| Layers | 6 | 12 | 4 | 6 |
| Hidden size | 256 | 256 | 64 | 384 |
| Heads | - | 6 | 4 | 6 |
| Voxel resolution | 24/32 | 24/32 | 24/32 | 24/32 |
| Point patch size | - | - | - | 32 |
| GPU device | NVIDIA A100 | NVIDIA A100 | NVIDIA A100 | NVIDIA A100 |

**Downstream Tasks Details**  Following Point-E [59], we use pre-trained PointNet++ as a classifier in all ACC, FID, and IS evaluations to extract the features and calculate the accuracy of generated point clouds. In point cloud generation and editing, we employ 8 or 4 steps for a parallel generation. The generation task utilizes initial voxel codebooks composed entirely of [MASK] tokens. In editing, we extract VQGAN features from the original point cloud to initialize the voxel codebooks. As for the transfer classification task on ScanObjectNN [83] and ModelNet40 [92], we fully follow the previous work [19, 60] configuration and trained 300 epochs with the AdamW optimizer, and used the voting strategy in testing.

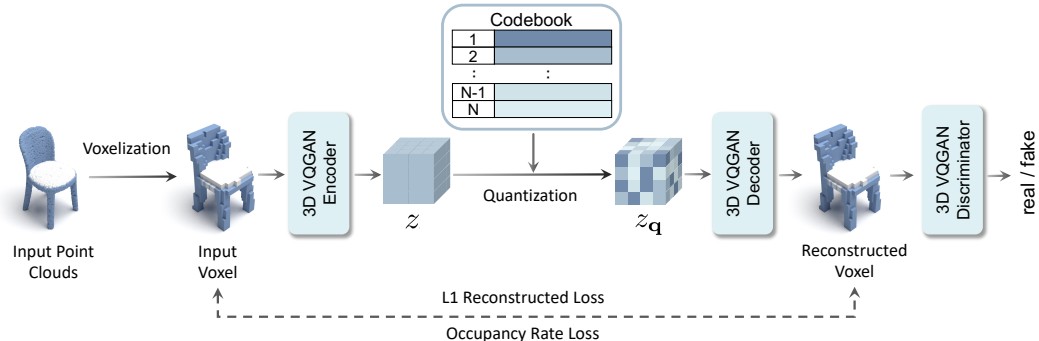

Figure 11: The training overview of 3D VQGAN. We introduce the occupancy rate loss to have a better reconstruction of the voxel.

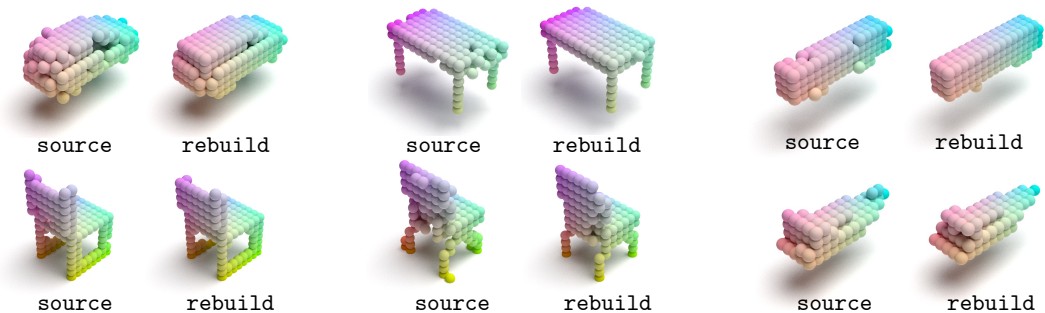

Figure 12: Reconstruction results of 3D VQGAN. Our model is capable of flawlessly reconstructing the original voxel, and can even rectify some noise.

## B.2 Implementation Details of 3D VQGAN

We show the detailed training overview of 3D VQGAN in Fig. 11. Following the training recipe of VQGAN [21], we use L1 loss to supervise the reconstruction of the voxel and feed the reconstructed voxel into the discriminator to ensure the generated authenticity by GAN loss. Besides the L1 loss and GAN loss, we also introduce the occupancy rate loss to make the occupancy rate of the reconstructed voxel grid similar to the ground truth so as to obtain a better reconstruction of the voxel. Fig. 12 illustrates the reconstructed results of 3D VQGAN, showcasing its impressive capabilities in the domain of 3D voxel reconstruction. As depicted in the figure, the model exhibits remarkable noise rectification capabilities. It not only reconstructs the voxels faithfully but also manages to rectify certain imperfections and artifacts present in the input data.

## C Additional Experiments

We conduct more experiments to further demonstrate the generation quality and universality of VPP. Including diversity & specific text-conditional generation, few-shot transfer classification, and ablation studies.

### C.1 Novel Categories Point Clouds Generation

Besides the generation of common categories, we also conduct novel categories generation experiments to show the generalization performance of VPP in Fig. 13. Based on the learning of common categories, VPP can generalize to generate more novel category shapes, which are visually reasonable, such as `a car boat`.

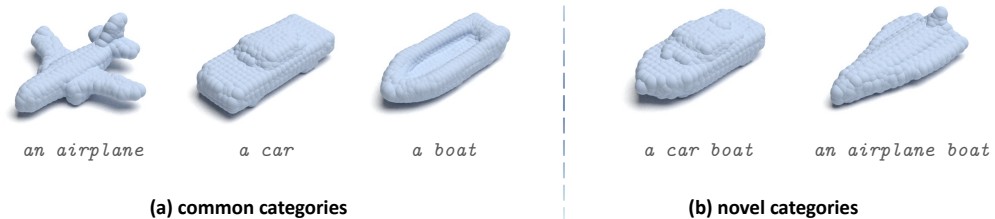

| an airplane | a car | a boat | | a car boat | an airplane boat |

**(a) common categories** | **(b) novel categories**

Figure 13: Novel categories point clouds generation.

Table 5: Downstream few-shot results on ModelNet40. Overall accuracy (%) w/o voting is reported.

| Method | 5-way | | 10-way | |
| --- | --- | --- | --- | --- |
| | 10-shot | 20-shot | 10-shot | 20-shot |
| *Supervised Learning Only* | | | | |
| DGCNN [89] | $31.6 \pm 2.8$ | $40.8 \pm 4.6$ | $19.9 \pm 2.1$ | $16.9 \pm 1.5$ |
| OcCo [86] | $90.6 \pm 2.8$ | $92.5 \pm 1.9$ | $82.9 \pm 1.3$ | $86.5 \pm 2.2$ |
| *with Self-Supervised Representation Learning* | | | | |
| Transformer [84] | $87.8 \pm 5.2$ | $93.3 \pm 4.3$ | $84.6 \pm 5.5$ | $89.4 \pm 6.3$ |
| OcCo [86] | $94.0 \pm 3.6$ | $95.9 \pm 2.3$ | $89.4 \pm 5.1$ | $92.4 \pm 4.6$ |
| Point-BERT [98] | $94.6 \pm 3.1$ | $96.3 \pm 2.7$ | $91.0 \pm 5.4$ | $92.7 \pm 5.1$ |
| MaskPoint [44] | $95.0 \pm 3.7$ | $97.2 \pm 1.7$ | $91.4 \pm 4.0$ | $93.4 \pm 3.5$ |
| Point-MAE [60] | $96.3 \pm 2.5$ | $97.8 \pm 1.8$ | $92.6 \pm 4.1$ | $95.0 \pm 3.0$ |
| Point-M2AE [102] | $96.8 \pm 1.8$ | $98.3 \pm 1.4$ | $92.3 \pm 4.5$ | $95.0 \pm 3.0$ |
| VPP (ours) | $\mathbf{96.9 \pm 1.9}$ | $\mathbf{98.3 \pm 1.5}$ | $\mathbf{93.0 \pm 4.0}$ | $\mathbf{95.4 \pm 3.1}$ |
| *with Pretrained Cross-Modal Teacher Representation Learning* | | | | |
| ACT [19] | $96.8 \pm 2.3$ | $98.0 \pm 1.4$ | $93.3 \pm 4.0$ | $95.6 \pm 2.8$ |
| I2P-MAE [103] | $97.0 \pm 1.8$ | $98.3 \pm 1.3$ | $92.6 \pm 5.0$ | $95.5 \pm 3.0$ |
| ReCon [68] | $97.3 \pm 1.9$ | $98.9 \pm 1.2$ | $93.3 \pm 3.9$ | $95.8 \pm 3.0$ |

## C.2 Few-Shot Transfer Classification

We conduct few-shot experiments on the ModelNet40 [92] dataset, and the results are shown in Table 5. In the self-supervised benchmark without the use of additional modality data, VPP achieves excellent performance compared to previous works.

## C.3 Ablation Study

**Training Hyper Parameter** Fig. 15 (a-b) shows the ablation study of the image-text features ratio and Gaussian noise strength. It can be observed that either too large or too small text-image feature ratio and noise are not conducive to the quality and diversity of 3D generation.

**Inference Parameters** To explore the parameter dependencies of the Mask Voxel Transformer in the inference process, we study the effects of temperature and iteration steps. The results are shown in Fig. 15 (c-d). It can be seen that moderate temperature achieves optimal generation classification accuracy. Higher temperature promotes model diversity, while with the increase of iteration steps, the model's FID initially decreases and then increases.

**Backbone Choice** Fig. 14 shows the selection of the CLIP model backbones during VPP training. Both ViT-B/32 and ViT/L14 achieve excellent accuracy and diversity.

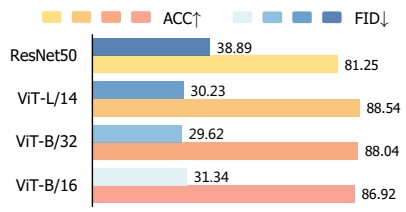

CLIP Backbone Choice

Figure 14: Ablation study on CLIP backbone choices.

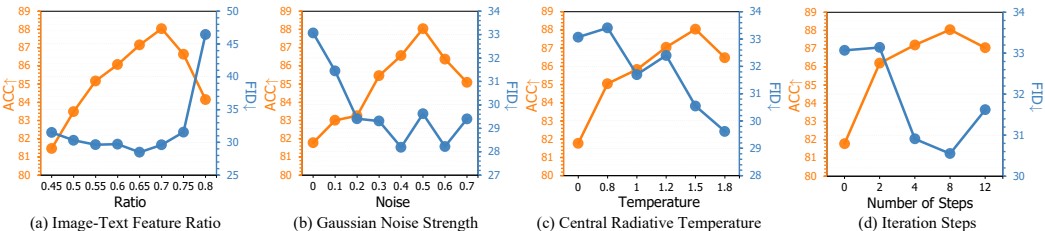

Figure 15: Ablation study on image-text feature ratio, Gaussian noise strength, inference temperature, and inference steps.

# D  Limitations & Future Works

Despite the substantial advantages of our model in terms of generation speed and applicability, there exists a considerable gap in generation quality compared to NeRF-based models [37, 65, 87]. To address this limitation, we intend to explore the utilization of larger models and more extensive datasets for training, *e.g.*, Objaverse [13, 14]. Furthermore, multi-modal large language models with combined comprehension and generation abilities [18] have advanced rapidly. Incorporating VPP's 3D generation capabilities into such models represents a potential avenue for future research.

## Broader Impact

VPP enables the rapid generation of high-quality 3D models in a fraction of a second. This has wide-ranging applications in fields such as gaming, virtual reality, augmented reality, and digital media production. These advances can lead to more immersive and visually stunning user experiences in entertainment and educational content. Furthermore, as a generative model, VPP may produce deceptive and malicious content and potentially impact associated employment opportunities.

