# OpenReview forum: "VPP: Efficient Conditional 3D Generation via Voxel-Point Progressive Representation"
_NeurIPS.cc/2023/Conference — NeurIPS 2023 poster_

### Official Review · Reviewer_U3Fo · 2023-07-05

**Soundness:** 3 good
**Presentation:** 3 good
**Contribution:** 3 good
**Rating:** 7
**Confidence:** 4

**Summary:**

The paper proposes an efficient conditional 3D generation via voxel-point progressive representation. More specifically, a voxel semantic generator and a point upsampler have been created to achieve efficient generation on multi-category objects. To verify the effectiveness of the method, extensive experiments with SOTA results are achieved.

**Strengths:**

1. The paper is well-written and well-organized.
2. Extensive experiments are conducted, and impressive results are obtained.

**Weaknesses:**

1. It seems that the paper can address point upsample task instead of upsample completion task, since upsample refers to generate dense points while completion refers to synthesize new points given partial scans.
2. It would be better to show some ablation studies on the network architectures such as removing or replacing specific components to see how they affect the performances.
3. It would be clearer to indicate Tab .3 in line 234 for the corresponding quantitative results.

**Questions:**

1. To reduce the degree of confidence when generating voxels because of the high masking rate, Gaussian noise is added to the prompt embedding.The question is instead of adding Gaussian noise, how about directly sampling smaller masking rate?
2. How is Eq. (2) obtained? It would be better to provide some hints on the calculation.

**Limitations:**

The authors have adequately addressed the limitations.

---

> ### Author Rebuttal · Authors · 2023-08-09
>
> Thank you for your valuable review.
>
> **W1: The capability of VPP for partial completion task.**
>
> Thank you for your suggestions, and we will make the necessary revisions to the pertinent statement. Furthermore, in **Figure 5** of the global response PDF, we show the partial generation results, which provide additional evidence to substantiate the efficacy of VPP.
>
> **W2: More ablation studies on the components of the proposed VPP.**
>
> - In **Figure 3** of the global response PDF, we present the visualization results of our proposed 3D VQGAN and Grid Smoother. It is observable that the VQGAN with $L_{occ}$ loss exhibits **enhanced reconstruction performance and adeptness in countering adversarial noise**. The Grid Smoother effectively accomplishes its role in voxel-point smoothing.
> - Furthermore, we quantitatively evaluate the impact of specific components on performance, as delineated in the following table.
>
> |          | Acc    | FID   | IS    |
> | -------- | ------ | ----- | ----- |
> | VPP w Locc   | 88.04% | 29.82 | 10.64 |
> | VPP w/o Locc | 84.19% | 35.75 | 9.98  |
>
> |                   | Acc    | FID   | IS    |
> | ----------------- | ------ | ----- | ----- |
> | VPP w Grid Smoother   | 88.04% | 29.82 | 10.64 |
> | VPP w/o Grid Smoother | 86.32%  | 32.13 | 10.25 |
>
> **W3: We have corrected all the typos.**
>
> **Q1: How about directly sampling a smaller masking rate instead of adding Gaussian noise?**
>
> Due to the cosine schedule generation steps during inference, we simulate various steps that may be encountered during inference by sampling from the arccos distribution. Therefore, the mask ratio during training is **not a constant value**, and it is necessary to sample high mask ratios as well. This is because the inference process starts with a 100% [mask token].
>
> **Q2: Some hints on the calculation of Eq.(2).**
>
> The approach of infusing adaptive Gaussian noise into cross-modal features originates from [1], wherein noise parameter λ directly influences the generated samples X to **modulate the generative dependence**. The authors derived and demonstrated this by employing the cumulative distribution function of the inner product of random vectors on a sphere, along with the utilization of the Gamma function.
>
>
>
> [1] Towards language-free training for text-to-image generation, CVPR 2022

---

> > ### Comment · Reviewer_U3Fo · 2023-08-14
> >
> > I would like to thank the authors for the clarification in the rebuttal, and my concerns are addressed.

---

> > > ### Author Response · Authors · 2023-08-14
> > > **Thanks for your recognition of our work**
> > >
> > > Dear Reviewer U3Fo,
> > >
> > > We sincerely appreciate your recognition of our work, and we're pleased that your concerns have been resolved! Your valuable suggestions significantly contributed to our work.

---

### Official Review · Reviewer_THuC · 2023-07-05

**Soundness:** 2 fair
**Presentation:** 3 good
**Contribution:** 2 fair
**Rating:** 4
**Confidence:** 4

**Summary:**

This work proposes to use a voxel-point progressive representation for efficient 3D generation, and it proposes a few architectures for different applications, including generation, editing, upsampling, and pre-training. Based on the reported results, the proposed method could generate various 3D shapes and could achieve competitive classification results.

**Strengths:**

1. The work proposes a voxel-point progressive representation, which could provide good 3D generation results on the ShapeNet dataset as shown in the reported experiments.

2. Many modules, such as 3D VQGAN, Voxel Semantic Generator, Grid Smoother, and Point Upsampler, have been proposed.

**Weaknesses:**

1. The proposed method could only generate 3D shapes belonging to the categories of the ShapeNet dataset.  It does not show any 3D shape results from unseen categories even with the help of CLIP.

2. Point-Voxel representation has been broadly studied in previous methods, such as [A-C], which has already been proven to be an efficient representation for 3D point cloud analysis.

3. The classification results for both the ScanObjectNN and ModelNet40 datasets are very saturated.  Also, the improvements seem to be very incremental.

[A] Liu, Z., Tang, H., Lin, Y., & Han, S. (2019). Point-voxel cnn for efficient 3d deep learning. Advances in Neural Information Processing Systems, 32.

[B] Zhang, C., Wan, H., Shen, X., & Wu, Z. (2022). PVT: Point‐voxel transformer for point cloud learning. International Journal of Intelligent Systems, 37(12), 11985-12008.

[C] Liu, Z., Tang, H., Zhao, S., Shao, K., & Han, S. (2021). Pvnas: 3d neural architecture search with point-voxel convolution. IEEE Transactions on Pattern Analysis and Machine Intelligence, 44(11), 8552-8568.

**Questions:**

1. Is the proposed method capable of generating 3D shapes of novel categories?

2. Is the proposed method capable of generating complete point clouds given a partial point cloud?

3. Does the proposed method demonstrate robustness when presented with noisy point clouds as input?

4. Is the generated 3D shapes caused by overfitting the ShapeNet dataset? Maybe the authors could try to retrieve the most similar shape in the ShapeNet dataset to prove its generation ability.

**Limitations:**

yes

---

> ### Author Rebuttal · Authors · 2023-08-09
>
> Thank you for your valuable review.
>
> **W1: Unseen categories generation.**
>
> - Differing from NeRF-based approaches like DreamFusion, although they can achieve open vocabulary zero-shot generation, the high computational time and training costs make practical utilization challenging. Our method strikes a balance among **multi-category generation, generation efficiency, and generation quality**, and is able to perform **multiple 3D downstream tasks** including conditional generation, editing, completion, and pre-training.
> - VPP trained on ShapeNet is incapable of achieving open vocabulary zero-shot generation, but it can produce novel categories to some extent. Furthermore, by employing a larger dataset Objaverse[1], VPP demonstrates the capability to generate more common objects. These results are illustrated in **Figure 2&6** of our global response PDF. Due to computational resource constraints, our model was trained for only 50 epochs on Objaverse. The current results represent preliminary findings.
>
> **W2: The correlation between the previous point-voxel representation and VPP.**
>
> - The previous point-voxel methods were predominantly architectural endeavors, aimed at achieving improved classification, detection, or segmentation performance.
> - VPP is the first endeavor that achieves efficient conditional 3D generation through the sharing of distinct representation advantages. We express our gratitude to the preceding point-voxel representation methodologies, and in the appendix, we will incorporate an Additional Related Work section to acknowledge and cite pertinent contributions.
>
> **W3: Improvement of classification results.**
>
> - Our Point Upsampler is implemented based on Point-MAE and achieved a 4.1 % performance improvement on ScanobjectNN. Our model achieves SOTA performance on the benchmark that only uses ShapeNet point clouds as pre-trained data.
> - Some recent work, such as I2P-MAE[2] and ReCon[3], achieved better performance through cross-modal tutors and cross-modal data.
>
> **Q1: Capability of generating 3D shapes of novel categories.**
>
> We conduct novel-categories generation experiments, which demonstrated the capability to generate novel-categories shapes, such as "a car boat". The results are presented in **Figure 6** of the global response PDF.
>
> **Q2: Capability of partial point cloud generation.**
>
> Thank you for your suggestion. We generate partial data by employing a block mask on the original point clouds. The results are depicted in **Figure 5** of the global response PDF, illustrating the partial generation capability of VPP. Furthermore, the generated samples exhibit diversity, providing further evidence of the performance of VPP.
>
> **Q3: Robustness of VPP when presented with noisy point clouds as input.**
>
> - As depicted in **Figure 3** of the global response PDF, our model demonstrates the capacity to not only faithfully reconstruct the provided input but also proficiently rectify distinct imperfections and noise inherent within the input data.
> - In addition, we quantitatively compare the robustness of the model to the training data. The table below shows the changes in text-conditional generation metrics after we added random scale, translate, and jitter disturbances to the input point cloud.
>
> |      | disturbance | Acc    | FID   | IS    |
> | ---- | ----------- | ------ | ----- | ----- |
> | VPP  | -           | 88.04% | 29.82 | 10.64 |
> | VPP  | random scale       | 87.63% | 31.20 | 10.15  |
> | VPP  | random translate   |83.76% | 36.92 | 8.71  |
> | VPP  | random jitter      | 87.29% | 30.68 | 10.44  |
>
>
> **Q4: Are the generated 3D shapes caused by overfitting the ShapeNet dataset?**
>
> Thank you for your suggestion. We conduct a retrieval evaluation on samples generated by VPP and the ShapeNet dataset. The results are shown in **Figure 8** of the global response PDF. There are no completely identical samples, therefore VPP is not overfitting the ShapeNet dataset. The results generated by VPP are more like an understanding and integration of shape knowledge.
>
>
>
> [1] Deitke M, Schwenk D, Salvador J, et al. Objaverse: A universe of annotated 3d objects. CVPR 2023
>
> [2] Zhang R, Wang L, Qiao Y, et al. Learning 3d representations from 2d pre-trained models via image-to-point masked autoencoders. CVPR 2023
>
> [3] Qi Z, Dong R, Fan G, et al. Contrast with Reconstruct: Contrastive 3D Representation Learning Guided by Generative Pretraining ICML 2023

---

> > ### Author Response · Authors · 2023-08-20
> > **Further Discussion**
> >
> > Dear Reviewer THuC,
> >
> > Thanks again for your valuable comments and suggestions! The Author-Reviewer discussion is coming to an end, and we hope that we have addressed all of your concerns. Please, let us know if you have any follow-up questions or concerns. We will be happy to answer them.
> >
> > Best Regards,
> >
> > Authors

---

### Official Review · Reviewer_YMLb · 2023-07-05

**Soundness:** 3 good
**Presentation:** 3 good
**Contribution:** 3 good
**Rating:** 6
**Confidence:** 3

**Summary:**

Authors propose an approach to generate 3D point clouds of objects with an image or text description as input. Authors use a pre-trained CLIP model to generate text/image embeddings and use this to first generate features in voxel space (Voxel Semantic Generator). These voxel features are then decoded into a coarse voxel grid. Authors then convert the voxels to point clouds and use a smoothing network to obtain a uniform point cloud. Authors use a transformer to then convert this point cloud into a detailed 3D shape (also point cloud).
Authors show qualitative and quantitative comparison with relevant baselines (CLIP-Sculptor, CLIP-Forge and Point-E).
Authors also show several applications like text and image conditioned 3D generation, shape editing and completion.

**Strengths:**

+ Ideas presented in the work are technically sound.
+ Presented results outperform competing baselines.
+ Authors submitted code. Although I did not run it but this is still appreciated. Code will also be released upon acceptance.
+ Paper is mostly well written. See comments below for minor improvements.

**Weaknesses:**

[Technical]
1. How is the proposed Voxel Semantic Generator different than “CLIP-Conditioned Coarse Transformer” from CLIP-Sculptor? They are essentially doing the same thing, 1. Learn a voxel based encoding of 3D shapes and 2. Use a transformer to learn how to unmask the 3D features conditioned on a prompt.
Can authors clarify this better?

2. L175-178: Since the GT for Grid Smoother is generated using furthest point sampling, why can’t we use FSP at inference time to smoothen the coarse voxel grid from the 3D VQGAN decoder?
Why do we need to learn a neural network?
How useful is adding KL loss over Chamfer loss?

3. L180: The output of the grid smoother is a point cloud (I assume this because it is trained with a Chamfer loss). L169 mentions that “point tokens” are masked which are then unmasked by the transformer. What exactly are the “point tokens”? Is it just the positional encoding of points?
Fig.2 (b) mentions “semantic tokens” what does this mean? It is not explained in Sec 3.3 (Point Upsampler).

4. Eq. 1: How useful is the occupancy loss over MSE? Is this critical? Please add an ablation study in supp. mat. to support the proposal.

[Minor]
- Fig. 2: Please add symbols used in the text to Fig.2. This makes it easier to follow the text and map various components of the pipeline.
- Add dimensionality of each embedding/latent code along with the symbols, eg: C_{pr} \in \mathbb{R}^{<whatever>\times<whatever>}. This significantly improves clarity and readability.
- L120: How is the mapping performed? How is the codebook generated? If an existing work is used, please add citation here? This is present in the supp. mat. to an extent. Add a pointer so that the reader knowns where to look.
- L140: Just curious, how important is the scheduling of the mask fraction?
L147: Is there significant performance difference at inference time between direct and multi-step prediction?
- L234: Table number is missing.
- Please avoid violent objects like guns for showing qualitative results. It is understandable that it is sometimes necessary to demonstrate performance but whenever possible, let us try to avoid this.

**Questions:**

Overall I'm positive about the work. There are some concerns about the novelty/necessity of some components (see above) and it would be great if authors can clarify these.

**Limitations:**

Discussion on Limitations and Broader Impact is not included in the main paper but it is present in the supp. mat. Authors are encouraged to add some discussion on limitations and future work in the main paper as it allows the community to better use and build upon your research.

---

> ### Author Rebuttal · Authors · 2023-08-09
>
> Thank you for your valuable review.
>
> **W1: Difference between Voxel Semantic Generator and CLIP-Sculptor?**
>
> - Our goal is to share the representation advantages of both voxels and points. The objective of the Voxel Semantic Generator is to provide positional encoding for the Point Upsampler. In contrast, CLIP-Sculptor entirely uses voxels as representation. The transformer sequence length is cubically proportional to the voxel resolution, resulting in slower inference speeds at high resolutions.
> - Additionally, VPP employs a **meticulously designed 3D VQGAN** to generate codebooks rather than VQVAE. VPP utilizes a mask ratio based on the **arccos distribution sampling** method, rather than a two-step unrolled training loss.
> - During inference, we use the **Central Radiative Temperature Schedule strategy** to get better performance.
>
> **W2: Why do we need the proposed Grid Smoother instead of FPS?**
>
> FPS cannot serve as a substitute for the proposed Grid Smoother module. Its objective is to **mitigate the representation gap** as possible. This is because the point cloud generated by the Voxel Semantic Generator is **discrete and grid**. Through a neural network, we obtain **continuous and uniformly distributed point clouds**. The effect of grid smoothing is demonstrated in **Figure 3** of the global response PDF. In contrast, the nature of FPS is downsampling, rendering it incapable of producing continuously valued points.
>
> Furthermore, FPS leads to a reduction in point quantity. Typically, our Voxel Generator can only produce between 200 to 500 points. Employing FPS downsampling would result in the loss of certain geometric information.
>
> **W3: The explanation of "point tokens" and "semantic tokens".**
>
> We employ a pre-trained Point-MAE encoder as the tokenizer of PointUpsampler to generate semantic point tokens. The point cloud generated by the grid smoother is utilized to provide positional encoding for these semantic tokens. We provide a more comprehensive explanation of this aspect in the revised version.
>
> **W4: How useful is the occupancy loss over MSE?**
>
> Thanks for your suggestion! We have incorporated this result into **Figure 3** of the global response PDF. As depicted, the addition of the $L_{occ}$ loss significantly improves the reconstruction performance compared to the vanilla VQGAN. This improvement not only faithfully restores the voxels but also effectively **corrects specific imperfections and noise** in the input data.
>
> Furthermore, the following table presents a quantitative performance comparison regarding the $L_{occ}$ loss.
>
> |              | Acc    | FID   | IS    |
> | ------------ | ------ | ----- | ----- |
> | VPP w Locc   | 88.04% | 29.82 | 10.64 |
> | VPP w/o Locc | 84.19% | 35.75 | 9.98  |
>
>
>
> **Minor**
>
> - We greatly appreciate all the constructive suggestions you have provided. We will incorporate symbols in the figures, add pointers, and correct typos, which will significantly enhance the readability.
> - Concerning the scheduling of the mask ratio, MaskGit[1] has conducted experiments on the scheduling strategy of mask fraction for parallel decoding. In this study, we adopt the optimal strategy.
> - In the ablation study Figure 8 of the main paper, we observed that a moderate number of inference steps (4/8) yields the best generative performance, while single-step generation leads to a decrease in accuracy by up to 6%.
>
> [1] Chang H, Zhang H, Jiang L, et al. Maskgit: Masked generative image transformer. CVPR 2022

---

> > ### Comment · Reviewer_YMLb · 2023-08-15
> > **Post rebuttal update**
> >
> > Thanks authors for the rebuttal. It addresses my concerns and I keep my positive rating of the work.

---

> > > ### Author Response · Authors · 2023-08-15
> > > **Thanks for your recognition of our work**
> > >
> > > Dear Reviewer YMLb,
> > >
> > > Thanks for keeping your positive rating of the work. Your constructive advice helps a lot to improve our work!

---

### Official Review · Reviewer_A5WF · 2023-07-06

**Soundness:** 3 good
**Presentation:** 2 fair
**Contribution:** 4 excellent
**Rating:** 5
**Confidence:** 5

**Summary:**

The VPP proposes a model for 3D generation. It utilizes both point-based and voxel-based representations. Voxel-based representations are used to generate the coarse tokens, and the point-based one further improves the result. Both of which are pretrained with MAE-like self-supervised method.

The proposed method applies to image-to-point, text-to-point, and point completion. The propsed method is novel and effective. However, some parts are not presented clearly.

**Strengths:**

1. The limitations of existing methods are well analyzed.
2. The proposed method is carefully designed and performs well.
3. The proposed method applies to more downstream tasks than exsiting methods.
4. The experiments show the effectiveness of the proposed method.

**Weaknesses:**

1. The inference efficiency in Table 1 only presents hours or seconds, it is recommended to provide the exact time.

Totally, the paper is not well presented. Some details are not shown clearly. To be specific,
2. The tokenizer for points upsampler is not described. What is it structure and how to train it?
3. It would be better to refer to some important symbols in Figure2. For example, prompt embedding in Figure 2 should also be marked as C_pr.
4. The GAN's structure and loss in 3D VQGAN are not introduced.
5. As described in Section 3.2, the mask voxel transformer will be forwarded multiple times for better quality. It should also be commented in Figure 2 or 3 for better presence.

Some typos:
6. In about line 151: "when the inputs of the mask voxel transformer are grid tokens with high masking fraction and prompt embedding we find that this will lead to the semantics of the predicted grid tokens being much more biased toward the prompt embedding". I think the authors may miss punctuations.

**Questions:**

1. I am not sure what parallel decoding in Figure 3 means.
2. When training the voxel semantic transformer, part of voxel tokens are masked, as shown in Figure 2a, and the masking ratio is about 64%. While all tokens are masked voxel tokens during inference. Is there a gap between training and inference?

---

> ### Author Rebuttal · Authors · 2023-08-09
>
> Thank you for your valuable review.
>
> **W: Some presentation suggestions. The unclear explanation of tokenizer and 3D VQGAN.**
>
> - We appreciate your suggestions regarding the presentation. In the revised version, we include **the exact time in Table 1**, incorporate **symbols** into the process diagram, and emphasize multi-step decoding in Figure 2&3 to enhance readability. Additionally, we correct all the typos.
>
> - About the tokenizer of the Point Upsampler, we employ a **pre-trained Point-MAE** encoder as the tokenizer to generate semantic point tokens. We will elaborate on it in the subsequent revised version.
>
> - The structural diagram of 3D VQGAN is shown in **Figure 4** of the global response PDF. We have described the proposed $L_{occ}$ loss in the main paper and demonstrated the training pipeline of 3D VQGAN in the supplementary materials. We will further enrich the explanation of the VQGAN loss in the revised version.
>
>
> **Q1: The meaning of parallel decoding.**
>
> Parallel decoding is introduced to distinguish it from sequential autoregressive decoding. In each step of parallel decoding, the model concurrently predicts all tokens in parallel, conditioned on the probability distribution from the previous step. Parallel decoding constitutes one of the crucial factors enabling VPP to achieve efficient generation.
>
> **Q2: Is there a gap between training and inference on the mask ratio?**
>
> VPP employs a cosine schedule during parallel decoding. Specifically, at each inference step:
> - The mask transformer predicts the logits of the codebook for all masked grid tokens.
> - Subsequently, we employ the cosine schedule to calculate the mask ratio at the current step and determine which tokens are to be fixed based on the logits scores.
> - The non-fixed tokens are replaced with mask tokens and proceed to the next step.
>
> Due to the cosine schedule of mask ratio during inference, we sample mask ratios from the arccos distribution during training to simulate various inference steps. The mean mask ratio is 0.64, yet it **doesn't mean a fixed mask ratio**.

---

> > ### Author Response · Authors · 2023-08-20
> > **Further Discussion**
> >
> > Dear Reviewer A5WF,
> >
> > Thanks again for your valuable comments and suggestions! The Author-Reviewer discussion is coming to an end, and we hope that we have addressed all of your concerns. Please, let us know if you have any follow-up questions or concerns. We will be happy to answer them.
> >
> > Best Regards,
> >
> > Authors

---

### Official Review · Reviewer_AZ71 · 2023-07-08

**Soundness:** 3 good
**Presentation:** 3 good
**Contribution:** 2 fair
**Rating:** 6
**Confidence:** 4

**Summary:**

The paper proposed a text-driven point cloud generation model that can be used for various downstream tasks such as generation, editing,
 completion and pretraining, while being very efficient. The method largely follows Muse [2], but adapted it to 3D point clouds. The model consists of multiple components, which are trained individually. First, a VQGAN is trained on the voxelized shapes, which embed a shape to a latent grid. Next, a masked transformer is trained on the latent grid to model its distribution, using CLIP feature as semantic conditioning. For the decoding stage, a grid smoother network moves the voxel centers to more homogeneous locations, while finally another set of point cloud VQVAE and masked transformer is trained to upsample the coarse point cloud to high resolution point cloud. The paper also demonstrated that the proposed components can be used in whole or in part to tackle a wide range of tasks.

**Strengths:**

* The proposed method is able to handle a wide range of tasks such as generation, editing, completion and pretraining.
* VPP significantly outperforms baseline methods on the main task of shape generation.
* The proposed method is computationally efficient compared to optimization-based methods such as DreamFusion.

**Weaknesses:**

* The components of the proposed method are not new -- they are mostly borrowed from previous works such as CLIP, MUSE and Point-MAE.
* The method contained a large number of stages, making it potentially difficult to implement or improve upon.
* Unlike diffusion-based methods such as DreamFusion, Dream3D, Magic3D, the proposed method does not generate surface or texture. It is also not clear if the proposed method is able to do zero-shot generation like these methods.
* The proposed method is only trained on ShapeNet, which is relatively small and less diverse compared to larger datasets such as Objaverse. It is not sure if the method is able to scale to larger datasets.

**Questions:**

* What is the benefit of using voxel as intermediate representation compared to directly encoding the point cloud?
* Missing table reference on L234.

**Limitations:**

Limitations and societal impacts are adequately addressed in the supplemental material.

---

> ### Author Rebuttal · Authors · 2023-08-09
>
> Thank you for your valuable review.
>
> **W1: Are the components of the proposed method new?**
>
> We have introduced some novel structures or strategies to adapt to 3D generation. Such as the **3D VQGAN with occ loss**, the **Grid Smoother** to smooth the gap between two representations, and the **Central Radiative Temperature Schedule** strategy during inference.
> Our goal is to share the merits of different representations for efficient multi-category generation. It is natural to draw inspiration from Muse for voxels and Point-MAE for points.
>
> **W2: Is VPP difficult to implement or improve upon?**
>
> - It is a common phenomenon to use three stages in the generative model. For example, Muse needs to train VQGAN, Base Transformer, and SuperRes Transformer before inference.
> - Furthermore, each module of VPP is based on CNN or Transformer, which is easy to implement. Modifying any stage could potentially contribute to improvements in the overall performance.
>
> **W3&4: Can VPP generate surface and do zero-shot generation? How about VPP trained on a larger dataset like Objaverse?**
>
> - First, we present the **surface reconstruction** results based on SAP in **Figure 1** of the PDF we provide in the Global Response. It demonstrates that VPP is capable of generating smooth surfaces.
> - Second, VPP trained on ShapeNet can not achieve open vocabulary zero-shot generation. But it can produce some **novel categories** to some extent. The results are shown in **Figure 6** of the provided PDF.
> - Third, by using a larger dataset **Objaverse**, VPP is capable of generating more common objects. We present the results in **Figure 2** of the provided PDF. Due to computational resource constraints, our model was trained for only 50 epochs on Objaverse. The current results represent preliminary findings.
>
> **Q1:  What is the benefit of using voxel as an intermediate representation compared to directly encoding the point cloud?**
>
> - We analyze the characteristics of different representations in the introduction. Due to significant shape differences across various categories, the structured and explicit positional information provides direct **spatial cues**, thereby aiding the generalization of multiple categories. This observation aligns with previous multi-category generation methods [1-2].
> - Additionally, point clouds possess continuous and sparse semantic information and heavily rely on positional encoding [3]. The point token represents **both local geometric semantics and positional information**, which significantly reduces the generation performance in complex multi-category scenarios. We show the results for directly encoding the point cloud in Figure 7 of the global response PDF. As observed, models trained without voxel representation exhibit inferior performance in generating object details compared to our current approach, particularly for generating objects with intricate structures, such as "an airplane".
>
> **Q2: We have corrected all the typos.**
>
> [1] Sanghi A, Chu H, Lambourne J G, et al. Clip-forge: Towards zero-shot text-to-shape generation. CVPR 2022
>
> [2] Sanghi A, Fu R, Liu V, et al. CLIP-Sculptor: Zero-Shot Generation of High-Fidelity and Diverse Shapes From Natural Language. CVPR 2023
>
> [3] Pang Y, Wang W, Tay F E H, et al. Masked autoencoders for point cloud self-supervised learning. ECCV 2022

---

> > ### Comment · Reviewer_AZ71 · 2023-08-14
> > **Thanks for the rebuttal**
> >
> > I would like to thank the authors for the clarification and the extra experiment on Objaverse. I have raised my rating as all of my concerns are resolved. The reason that prevents me from giving any higher rating is that, despite the model seems to be very powerful, its zero-shot generation capability appears to be very limited, which will limit its usefulness.

---

> > > ### Author Response · Authors · 2023-08-14
> > > **Thanks for your recognition of our work**
> > >
> > > Dear Reviewer AZ71,
> > >
> > > We sincerely appreciate your recognition of our work and insightful comments. We will continue to explore training on large datasets to improve the zero-shot capability of VPP and include all new discussions and results in the revised version.

---

### Author Rebuttal · Authors · 2023-08-09

We sincerely thank all reviewers for your valuable feedback that significantly contributed to our work. VPP achieves efficient, multi-category, high-quality conditional generation through voxel-point progressive representation, and is capable of performing various tasks such as editing, completion, and pre-training.

We present additional experimental results in the following PDF to offer a more intuitive and clear response to the reviewer's relevant comments and questions. Moreover, we further demonstrate the optimized mesh generation performance and the generation results using a larger dataset. Specifically, this PDF includes the following:

Figure 1: Improved surface reconstruction results through SAP.

Figure 2: Visualization of text-conditioned generation results using Objaverse dataset.

Figure 3: Visualization results of 3D VQGAN and Grid Smoother.

Figure 4: Structure diagram of the 3D VQGAN.

Figure 5: Generation results of partial inputs.

Figure 6: Novel categories generation results.

Figure 7: Ablation experiment that only uses point clouds as representation.

Figure 8: Ablation experiment that retrieval ShapeNet samples.

Due to space limitations, we employ a two-column layout.

---

### Decision · Program_Chairs · 2023-09-21

**Decision:**

Accept (poster)

**Comment:**

This paper proposes a conditional 3D generation method via voxel-point progressive representation. It receives two weak accepts, one borderline accept, one accept and one borderline reject. The authors well resolved the reviewers' concerns. Reviewer THuC raises two new concerns during discussion. However, it is pretty late for the author to response. The AC thinks the new issues do not affect the contribution of this paper and recommends acceptance for this paper. The authors are urged to solve the new concerns in the camera ready version.